# Examining the impact of maternal experiences of domestic violence on the mental health of their adolescent children in India

Amritha Gourisankar[1,2]*, Preethi Ravi[1,2], Ameeta Kalokhe[1,2], Rachel Waford Hall[2],
Nilakshi Vaidya[3,4], Eesha Sharma[5], Bharath Holla[6], Debasish Basu[7], Rose Dawn Bharath[8],
Amit Chakrabarti[9], Sylvane Desrivieres[10], Matthew Hickman[11], Kamakshi Kartik[12],
Krishnaveni Ghattu[13], Kumaran Kalyanaraman[13], Murali Krishna[14], Rebecca Kuriyan[15],
Pratima Murthy[6], Dimitri Papadopoulos Orfanos[16], Meera Purushottam[17],
Sunita Simon Kurpad[18], Rajkumar Lenin Singh[7], Roshan Lourembam Singh[7],
Bhagyalakshmi Nanjayya Subodh[19], Mireille B. Toledano[20], Vivek Benegal[3],
Gunter Schumann[4,21], Kartik Kalyanram[12], the cVEDA Consortium¶

1 Emory University School of Medicine, Atlanta, United States of America, 2 Emory Rollins School of Public Health, Atlanta, United States of America, 3 Centre for Addiction Medicine, National Institute of Mental Health and Neurosciences, Bangalore, India, 4 Centre for Population Neuroscience and Precision Medicine (PONS), Department of Psychiatry and Neuroscience, Charité Universitätsmedizin Berlin, Berlin, Germany, 5 Department of Child and Adolescent Psychiatry, National Institute of Mental Health and Neurosciences, Bangalore, India, 6 Department of Psychiatry, National Institute of Mental Health and Neurosciences, Bangalore, India, 7 Department of Psychiatry, Regional Institute of Medical Sciences, Imphal, Manipur, India, 8 Department of Neuroimaging and Interventional Radiology, National Institute of Mental Health and Neurosciences, Bangalore, India, 9 Centre for Ageing and Mental Health, Indian Council of Medical Research (ICMR), Kolkata, India, 10 Centre for Population Neuroscience and Precision Medicine, MRC Social, Genetic, Developmental Psychiatry Centre, Institute of Psychology, Psychiatry & Neuroscience, King's College London, London, United Kingdom, 11 Bristol Medical School, University of Bristol, Bristol, United Kingdom, 12 Rishi Valley Rural Health Center, Madanapalle, India, 13 Epidemiology Research Unit, CSI Holdsworth Memorial Hospital, Mysore, India, 14 Institute of Public Health, Bangalore, India, 15 NeuroSpin, CEA, Université Paris-Saclay, Paris, France, 16 Molecular Genetics Laboratory, National Institute of Mental Health and Neurosciences, Bangalore, India, 17 Division of Nutrition, St. John's Research Institute, Bangalore, India, 18 Department of Psychiatry, Postgraduate Institute of Medical Education and Research, Chandigarh, India, 19 Medical Research Council Centre for Environment and Health, School of Public Health, Imperial College London, London, United Kingdom, 20 Institute for Science and Technology of Brain-inspired Intelligence (ISTBI), Fudan University, Shanghai, China, 21 Department of Psychiatry & Department of Medical Ethics, St. John's Medical College and Hospital, Bangalore, India

¶ Membership of the cVEDA Consortium listed in the Acknowledgements.
* gouri012@umn.edu

## Abstract

### Background

Domestic violence (DV) is experienced by one in three women in India and is linked to poor mental health outcomes. We hypothesize that maternal experiences of DV can have negative impacts on the mental health of their children. Previous studies have demonstrated this link in Western countries, however culturally specific manifestations of DV and mental health disorders and socio-cultural differences in

**Data availability statement:** The dataset generated during the cVEDA study are available to interested researchers as per the cVEDA data sharing guidelines. The data is owned by the cVEDA study team and we do not have permission to share the data https://cveda-project.org/access-to-the-c-veda-dataset/.

**Funding:** The author(s) received no specific funding for this work.

**Competing interests:** The authors have declared that no competing interests exist.

parent-child relationships and home environments necessitate deeper understanding of the impacts of maternal experiences of DV on children in the Indian context.

## Methods

This study presents a secondary analysis of data collected from a seven-center study in urban and rural India examining mental health disorders among adolescents aged 12–17 years and psychological, physical, and sexual abuse affecting their mothers. The Indian Family Violence and Control Scale (IFVCS) was used to examine experiences of DV among mothers and the Mini International Neuropsychiatric Interview– Kid (MINI-Kid) was used to examine mental health outcomes among adolescents. Multivariate analyses examined the associations between maternal DV and adolescent mental disorders.

## Results

Data from 2,784 adolescent-mother pairs were analyzed. In bivariate analyses, maternal experiences of physical, psychological, and sexual abuse were significantly associated with adolescent common mental disorders including anxiety and depression (p < 0.05). After adjusting for adolescent gender, site, and education status in the multivariate analysis, physical, sexual, and any DV were significantly associated with adolescent anxiety disorders and common mental disorders. Physical abuse was significantly associated with adolescent depressive disorders.

## Conclusions

These results suggest that exposure to maternal DV significantly impacts adolescent mental health in India and underscore the need to develop trauma-informed school programs and enhance DV prevention for women in India.

---

## Introduction

Domestic violence (DV), including physical, sexual, and emotional violence, is experienced by approximately 1 in 3 women in India [1]. Women in India who experience DV are more likely to be diagnosed with anxiety, depression, PTSD, suicidality, or fall in the broader category of having a common mental disorder (CMD), which includes anxiety and depressive disorders [2,3]. CMDs among women experiencing DV are associated with greater reporting of gynecological symptoms such as dyspareunia or menstrual cramps [4]. Studies in India also indicate that lifetime experiences of DV are associated with adverse reproductive outcomes including miscarriage, stillbirth, preterm birth, and complications during labor [5,6]. Associations between maternal experiences of DV and adverse health outcomes in children, including malnutrition, stunted growth, and asthma, have been demonstrated in various contexts including in the United States and Bangladesh [7–9]. They have also been linked to emotional, behavioral, and academic difficulties among children across India specifically [10,11].

Despite the aforementioned associations, however, there remain gaps in knowledge related to the impact of maternal experiences of DV on the mental health outcomes of children, including common mental disorders (e.g., depression, anxiety), externalizing disorders (e.g., attention deficit/hyperactivity disorder), and suicidality, in the Indian context.

Evidence from other countries suggests that maternal experiences of DV are associated with poor mental health outcomes among children. In the United States, for example, Fredland et al. demonstrated associations between maternal experiences of DV and poor maternal mental health; poor maternal mental health, in turn, predicted symptoms of depression and aggression in children [12]. Children exposed to maternal experiences of DV are also more likely to demonstrate symptoms characteristic of conduct disorders, in addition to clinically significant internalizing and externalizing behaviors [13–16]. In multiple Western countries, exposure to maternal experiences of DV during childhood has been shown to be associated with the development of post-traumatic stress disorder (PTSD), oppositional defiant disorder (ODD), anxiety disorders, and major depressive disorder (MDD) in adolescence and young adulthood [17–20]. These findings have also been replicated in multiple low and middle-income countries, where exposure to DV is typically associated with even higher prevalences of suicidal behavior, substance use, and mental disorders including PTSD and MDD in children [21].

In the Indian context, the impact of maternal experiences of DV on the development of mental disorders in children warrants further investigation. In India, for example, joint family structures (wherein women often live with their husbands' parents and other family members after marriage) are pervasive. Such family structures enable members of the in-law families to serve as key support systems for women, but also enable perpetration of DV either directly or indirectly via influence on a male spouse. Other manifestations of DV specific to the Indian context include perpetration of emotional violence by forcibly sending women to their parental homes, perpetration of physical violence through the use of readily available items (e.g., stones, chemicals that cause burns), and the withholding of contraception until male children are born as a means of control [22–25]. Mental health disorders may also present in culturally specific ways, including the presence of somatic symptoms related to CMDs among Indian women [4].

Adolescence is a critical developmental period during which milestones related to cognition, social learning, and personality are achieved, and during which psychopathology may manifest [26,27]. Conduct disorders and ADHD typically peak during adolescence, and depression and anxiety often first manifest at this stage [28,29]. Adolescence may also be a period during which individuals are vulnerable to the effects of exposure to maternal DV, among other adverse childhood experiences (ACEs). The effects of ACEs broadly have been associated with chronic diseases and psychological disorders that may present in childhood, adolescence, or later adulthood [30,31]. Support services for children exposed to domestic violence (e.g., school-based interventions) may be effective in reducing mental health disorders, however, these services may be limited in certain regions, including rural parts of India [32,33].

Therefore, we proposed to examine the association between experiences of maternal DV and mental disorders among paired adolescent children in India. We hypothesized that experiences of maternal DV would be associated with increased mental health symptoms among corresponding adolescents. Given the high burden of DV on women of childbearing age in India, in addition to the prevalence of mental disorders among children, characterizing this relationship is especially critical. The identification of the type of DV exposure on specific mental health disorders among adolescents can be utilized to design culturally-tailored screening and preventive mental health interventions for adolescents exposed to maternal DV and help prioritize DV prevention efforts among mothers.

## Methods

### Introduction

This study employs secondary analysis of data collected from the multi-center study, *Consortium on Vulnerability to Externalizing Disorders and Addiction (cVEDA), India* (https://cveda.org). The cVEDA study was a longitudinal cohort study of mothers and children across seven health facilities in India that aimed to identify risk and resilience factors affecting

the development of mental health disorders in children, adolescents, and young adults. The study utilized an observational cohort design with data collection at baseline, in addition to follow-up data collection 1 and 2 years later for specific measures.

The secondary data analysis was conducted by two study team members (AS, PR) and involves a subset of the cVEDA cohort comprised of all children aged 12–17 years enrolled in the study at the time of baseline data collection (also known as the C2 cohort). The C2 cohort was specifically selected as children 12–17 were determined to be most likely to witness and understand household violence, compared with the C1 cohort comprised of children aged 6–12 who may not understand specific dimensions of household violence (e.g., psychological abuse) and the C3 cohort comprised of young adults aged 18–23 who may not be living in the household, and therefore may not be exposed to household violence.

## Population and sample

The study was conducted at seven culturally, socioeconomically, and geographically diverse sites across six states in India and utilized convenience sampling to identify mother-child pairs. More information on the sites can be found in the cVEDA study protocol [34]. Recruitment utilized local schools and hospitals as gatekeepers, in addition to snowballing techniques to capitalize upon the networks of participants. The study included a battery of questionnaires related to sociodemographic information, environmental exposures, psychosocial and personality characteristics, school climate, technology use, food habits, temperament, adverse experiences of the child, medical history of the child and family, pregnancy history of the mother, and experiences of DV of the mother. Additional details related to study sampling techniques and study population at each site can be found in the published cVEDA study protocol [34].

Although the analysis presented here is limited to children aged 12–17, resulting in a sample size of 2,784 mother-child pairs, the inclusion criteria for the parent study included: parents with children aged 6–23 years, who provided informed consent and were able to complete the necessary verbal assessments and neuropsychiatric testing. Exclusion criteria included inability of adults and/or children to complete assessments, and children with legal blindness or deafness, active seizure disorder, or severe physical or active mental illness. Only one child per family was enrolled in the study.

## Instruments

The primary predictor, maternal experiences of DV, was measured using the Indian Family Violence and Control Scale, a culturally specific 63-item instrument which is used to examine experiences of control and physical, psychological, and sexual abuse among married Indian women by their husbands and families-in law [35]. The primary outcomes, common mental health disorders in the paired children, were assessed using the Mini-International Neuropsychiatric Interview for Kids, or MINI-Kid. This instrument screens for psychiatric disorders as defined by the DSM-IV, in children and adolescents aged 6–17 years [36]. The psychiatric disorders analyzed in the study, screened for by MINI-Kid include: depressive disorders (i.e., major depressive disorder (MDD) and dysthymia), anxiety disorders (i.e., social phobia, specific phobias, panic disorder, generalized anxiety disorder (GAD), and post-traumatic stress disorder (PTSD)), suicidal ideation, and externalizing disorders (i.e., attention deficit-hyperactivity disorder (ADHD), conduct disorder, and oppositional defiant disorder (ODD)). The MINI-Kid also screened for substance use disorder, obsessive-compulsive disorder (OCD), and psychotic disorders including schizophrenia.

We examined adverse child events and demographics as potential confounders. The Adverse Childhood Experiences – International Questionnaire (ACE-IQ) is a survey designed to measure the number and types of experiences related to abuse and household dysfunction (e.g., parental drug use) that children are exposed to in their household and/or community. Based on a version developed in the United States, the International Questionnaire aims to characterize ACEs across on a global scale, and includes questions related to household violence (e.g., yelling, beating) and broader, community-level violence (e.g., military actions, forced migration) [37]. The Socio-Demographic Information and Migration Questionnaire (SDI-M) was adapted from the National Family Health Survey section on household information, and asked

specifically about participants' religion, caste, gender, and educational level, household size and characteristics [1]. The SDI-M also contains questions about members of the household who have migrated and the nature of their work. The Wealth Index was also utilized as a demographic metric. It was developed by the United States Agency for International Development (USAID) Demographic and Health Surveys (DHS) Program, and yields data which is reported in quintiles based on land ownership, ownership of other assets, household construction materials, availability of utilities such as toilets, and other country-specific markers [38].

## Data collection

During the data collection process for the cVEDA study, surveys were conducted by trained research staff through private face-to-face interviews with mother and child pairs at each study site. The SDI-M, Wealth Index, and MINI-Kid were administered directly to the child, with the mother present, while the DV questionnaire was administered directly to the mother alone. The ACE-IQ was administered directly to the child alone. The staff members administering the questionnaires were primarily psychologists, along with staff members with certifications in nursing or other health professions. The surveys were translated into the corresponding regional languages and administered in the primary language of the participant. Access to the cVEDA dataset for secondary analysis was obtained following the corresponding approval process. Additional information on the survey process and ethics clearance can be found in the cVEDA study protocol [34].

## Ethics declaration

The cVEDA study received clearance from the Health Ministry's Screening Committee, Ministry of Health and Family Welfare, Government of India, and ethics approvals at all participating centres in India and in the UK. The study also had an internal ethics advisory board that reviewed any ethical issues that arise and supported recruitment centres in their operations. Participants were recruited after written informed consent. In the case of minors (<18 years of age), informed consent was taken from the legal guardian (usually parents) and assent from the child/adolescent. The secondary data analysis resulting in this manuscript was waived by the Emory University IRB committee, as no new data collection was conducted.

## Data analysis

The data was collected using Delosis Psytools (https://www.delosis.com/) software, transferred to Microsoft Excel for de-identification and cleaning, and then SPSS Statistics software for data analysis.

## Violence scale

Three exposure variables were created from each of the subdomains of the IFVCS to measure physical abuse, psychological abuse, and sexual abuse. A fourth subdomain of control was not included in this study due to difficulty in assessing how control-related factors were perceived by children in the household. Each variable had five response options, "Never," "About Once or Twice in the Past Year," "About Once a Month," "About Once a Week," and "Not in the Past Year but Previously in My Married Life."

To calculate past-year DV, each variable was analyzed categorically and defined as present if mothers responded affirmatively to *any* of the items within the subscale by endorsing "About Once or Twice in the Past Year," "About Once a Month," or "About Once a Week," and absent if mothers either denied experiencing all items in the subscale, or only endorsed "Not in the Past Year but Previously in My Married Life."

## Mental health scale

Five outcome variables were created from the MINI-Kid: 1) depressive disorders, defined as the presence of either major depressive disorder or dysthymia [39], 2) anxiety disorders, defined by the DSM-V and supporting literature to encompass

social phobia, specific phobias, panic disorder, generalized anxiety disorder (GAD), and post-traumatic stress disorder (PTSD) [40,41], and 3) common mental disorders, defined as the combined category of anxiety and depressive disorders [39]. Each of these three outcome variables was initially analyzed categorically in the bivariate analysis and coded as '0' for not meeting the screening criteria for *any* of the included disorders in that category, and '1' for meeting the screening criteria for *at least one* of the included disorders. These five outcome variables were then analyzed as continuous variables, coded based on the range of their score, starting at '0' for not meeting the screening criteria for *any* of the included disorders in that category, and increasing in value depending on the number of disorders they met screening criteria for. The scaled (continuous) variables were ultimately utilized in the bivariate and multivariate results tables, as they better represented the full range and variability of outcomes.

## Descriptive statistics

Descriptive statistics were first used to examine the demographic characteristics and the distribution of the exposure (maternal experience of DV) and outcome variables (presence of mental health disorder in child).

## Bivariate analysis

Bivariate analyses utilizing the Chi-square test for independence were conducted to describe the associations between the categorical exposure and outcome variables. A two-tailed alpha of 0.05 was deemed significant for the Chi-square test. The exposure variables were the violence scale variables of psychological, physical, and sexual abuse, and any DV, coded as no or yes ('0' or '1'). The outcome variables were the mental health disorder categories described above, also coded as no or yes.

## Confounding analysis

Next, a confounding analysis was conducted to identify which of the following demographic variables among the children from the SDI-M (age, household size, education level (less than primary school or greater than primary school), site, gender, location (rural or urban), slum dwelling (no or yes), reserved caste (no or yes), joint family (no or yes), education dropout (no or yes)) and quintile of the household from the Wealth Index were independently associated with both the exposure variables (maternal DV subtypes (physical, psychological, sexual) and any DV) and the outcome variables (mental health conditions of anxiety, depression, and common mental disorders). Additionally, ACEs variables (ACE-IQ binary and ACE-IQ frequency), as measured by the ACE-IQ, were analyzed to determine if they were confounders of the exposure and outcome variables. Binary and frequency refer to two methods of scoring the ACE-IQ tool: the frequency scoring accounts for differences in how often an adverse experience occurred, whereas the binary scoring accounts for whether an adverse experience occurred. Adversity frequency better captured the variability yielded by the ACE-IQ and was therefore utilized in our analyses. In analyses involving the ACE-IQ, we removed three questions that asked about witnessing parental experiences of domestic violence (F6-F8 on the questionnaire) from composite scoring as they were redundant with our exposure variables (from the IFVCS) of parental experiences of DV, and would therefore be redundant and illogical to test for as confounders.

The variables that were significantly associated (p < 0.05) with both an exposure variable and an outcome variable were identified as confounders. The identified confounders were gender, site, education dropout (no or yes), adversity binary, and adversity frequency.

## Multivariate analysis

Lastly, a multivariate analysis using logistic regression was used to examine the association between the various forms of maternal DV experience and anxiety, depression, and common mental disorders, after controlling for ACEs and

demographic variables identified as potential confounders. Recognizing that ACEs could be potential confounders or part of a causal pathway, regression models including and excluding ACEs as covariates were run.

## Results

### Description of results

Data from 2,784 mother-child pairs were utilized in our analysis, with 11.3% recruited from PGIMER, Union Territory, 8.0% from RVRHC, Andhra Pradesh, 33.8% from NIMHANS, Karnataka, 20.2% from SJRI, Karnataka, 14.4% from RIMS, Manipur, 0.7% from CSIHMS, Karnataka, and 11.5% from ICMR-NIOH, West Bengal (see Table 1). The average age of the children was 14 years ($\sigma = 1.5$ years), and 47% of child participants were female. The average household size was 5 persons ($\sigma = 2.6$ persons), with 19.1% residing in joint families. Approximately one-third (31.5%) resided in rural areas and 11.1% in slum settlements. 43.8% of the participants were from reserved castes, 5% had dropped out or temporarily discontinued their education, and 1.1% were below a primary school level.

Among the adolescents, the prevalence of anxiety disorders was 5.3%, depressive disorders was 3.2%, and CMDs was 7.4% (see Table 2). Among the mothers, the prevalence of any type of DV, in the past year was 36.8% (or 947/2570). The prevalence of past-year physical abuse was 14.5%, past-year psychological abuse was 36.0%, and past-year sexual abuse was 3.9%.

Past-year physical and sexual abuse, but not psychological abuse nor any DV, were significantly associated with anxiety disorders in the paired children (see Table 3). Adolescent depressive disorders were significantly associated with past-year physical, psychological, and sexual abuse, and any maternal DV experience. Additionally, adolescent common mental disorders were significantly associated with past-year maternal physical and sexual abuse, and any DV.

Confounders were identified by conducting bivariate analyses of sociodemographic variables (obtained from the SDI-M), with each of the MINI-Kid and the IFVCS variables. Those that were significantly associated with IFVCS and MINI-kid variables are shown below (see Table 4). Additionally, any DV was associated with adversity binary and adversity frequency, both of which report outcomes of the ACE-IQ. Adversity binary and frequency were associated with any DV, physical, psychological, and sexual abuse (see Table 5). Both adversity binary and adversity frequency were associated with anxiety disorders, depressive disorders, and common mental disorders (see Table 5). As a result, adversity binary and adversity frequency were determined to be confounders. With regards to Wealth Index, there is no significant association between Wealth Index quintile and any DV, physical abuse, psychological abuse, or sexual abuse. There is, however, an association between Wealth Index score and both physical and psychological abuse (see Table 6). There is no association between Wealth Index quintile or Wealth Index score and anxiety, depression, or common mental disorders (see Table 6). It is reasonable, therefore, to label Wealth Index score as a confounding variable.

### Multivariate analysis including confounders

After determining specific confounders, bivariate analysis demonstrated no significant association between any DV and anxiety disorders. After controlling for site, education status, and gender with or without adversity frequency, maternal DV experience was not significantly associated with anxiety disorder in the paired adolescent (see Table 7).

Bivariate analysis demonstrated a significant association between any DV and depressive disorders. After controlling for site, education status, and gender, maternal DV experience was significantly associated with depressive disorders. This association did not hold when adversity frequency was added to the model (see Table 8).

Bivariate analysis demonstrated a significant association between common mental disorders and any DV, but after controlling for site, education status, and gender, maternal DV was no longer associated with CMDs, with or without adversity frequency included in the analysis. There are significant associations between common mental disorders and site (without inclusion of adversity frequency only), education status (with or without inclusion of adversity frequency), gender (with and without inclusion of adversity frequency), and adversity frequency in the multivariate analysis (see Table 9).

**Table 1. Descriptive statistics about adolescent participants.**

**N = 2784**

| | Mean (SD) |
|---|---|
| Age (years) | 14 (1.5) |
| Household Size | 5.1 (2.6) |
| | **No.** |
| Gender | |
| Female | 1290 |
| Location | |
| Rural | 863 |
| Residents in Slum Settlements | |
| Yes | 304 |
| Caste | |
| Reserved castes | 1174 |
| Family structure | |
| Joint | 523 |
| Education status | |
| Dropped out of school | 138 |
| Educational level | |
| Below primary | 30 |
| Primary | 204 |
| Upper primary/Middle | 1225 |
| Secondary | 955 |
| Higher secondary and Above | 328 |
| Site of Enrollment | |
| PGIMER (1), Union Territory | 314 |
| RVRHC (2), Andhra Pradesh | 222 |
| NIMHANS (3), Karnataka | 942 |
| SJRI (4), Karnataka | 563 |
| RIMS, IMPHAL (5), Manipur | 402 |
| CSIHMS, MYSORE (6), Karnataka | 20 |
| ICMR-NIOH, KOLKATA (7), West Bengal | 321 |
| Wealth Index | **N = 3169, (%)** |
| Lowest Quintile | 564, 12.7 |
| Quintile 2 | 737, 16.6 |
| Quintile 3 | 681, 15.3 |
| Quintile 4 | 580, 13.1 |
| Highest Quintile | 607, 13.7 |

Analyses controlling for site, education dropout status and gender with or without adversity frequency for the association between each of the subtypes of DV and mental disorders are listed in the appendix. In multivariate analyses, physical abuse was significantly correlated with anxiety disorders (OR = 1.8), depressive disorders (OR = 2.3), and common mental disorders (OR = 2.1), when adversity frequency was not included as a confounder. All these associations were no longer significant when adversity frequency was included in the analysis as a confounder. Additionally, sexual abuse was significantly associated with anxiety disorders (OR = 2.1) and common mental disorders (OR = 1.9) when adversity frequency was not included as a confounder; however, these associations were no longer significant when adversity

**Table 2. Prevalence of anxiety, depression, and common mental disorders among adolescents and prevalence of maternal exposure to any dv, physical abuse, psychological abuse, and sexual abuse.**

| Adolescent outcome (N=4437) | |
|---|---|
| Anxiety disorders | 5.4% (147) |
| Depressive disorders | 3.5% (96) |
| Common mental disorders | 7.5 (206) |
| Maternal exposure (N=4437) | |
| Any DV | 36.8% (947) |
| Physical abuse | 14.5% (389) |
| Psychological abuse | 36.0% (965) |
| Sexual abuse | 3.9% (100) |

frequency was included as a confounder. In all analyses, site, education dropout status, and gender were included as

**Table 3. Bivariate analyses of anxiety, depressive, and common mental disorders with past-year DV.**

| | Anxiety disorders | | | Depressive disorders | | | Common mental disorders | | |
|---|---|---|---|---|---|---|---|---|---|
| | No N=2393 | Yes N=132 | P-Value | No N=2433 | Yes N=92 | P-Value | No N=2336 | Yes N=189 | P-Value |
| Any DV | 864 (36.1%) | 56 (42.4%) | .142 | 871 (35.8%) | 49 (53.3%) | **0.001** | 835 (35.7%) | 85 (45.0%) | **0.011** |
| Physical Abuse | 352 (14.2%) | 33 (23.9%) | **0.002** | 356 (14.1%) | 29 (31.2%) | **0.000** | 332 (13.7%) | 53 (27.0%) | **0.000** |
| Psychological Abuse | 882 (35.4%) | 52 (38.2%) | 0.504 | 892 (35.2%) | 42 (45.7%) | **0.040** | 856 (35.2%) | 78 (40.4%) | 0.144 |
| Sexual Abuse | 88 (3.7%) | 11 (8.2%) | **0.008** | 90 (3.7%) | 9 (9.7%) | **0.003** | 84 (3.6%) | 15 (7.8%) | **0.004** |

**Table 4. Confounding analyses of SDI variables with exposure and outcome variables.**

| | Physical violence | Psychological violence | Sexual violence | Any DV | Depressive disorders | Anxiety disorders | Common mental disorders |
|---|---|---|---|---|---|---|---|
| Age | .246 | .794 | .775 | .643 | .754 | .451 | .553 |
| Household size | **.004** | **.000** | .069 | **.000** | .792 | .679 | .067 |
| Education group | .179 | .702 | .303 | .848 | **.010** | .988 | .913 |
| Site | **.000** | **.000** | **.000** | **.000** | **.000** | **.000** | **.000** |
| Gender | .868 | **.000** | .096 | **.000** | **.008** | .055 | **.008** |
| Rural (Y/N) | .377 | **.000** | .273 | **.003** | .815 | .167 | .085 |
| Slum dwelling (Y/N) | **.004** | **.000** | .130 | **.000** | .061 | .964 | .261 |
| Reserved caste (Y/N) | .066 | .654 | .536 | .430 | .690 | .660 | .402 |
| Joint Family (Y/N) | .089 | **.000** | .372 | **.002** | .562 | .355 | .165 |
| Education dropout (Y/N) | **.000** | **.000** | **.000** | **.000** | **.000** | .263 | **.000** |

confounders. Gender was significantly associated with exposure and outcome variables in all regression analyses. With regards to the associations between physical abuse and common mental disorders, anxiety disorders; sexual abuse and common mental disorders, anxiety disorders, the site, education dropout status and gender confounders were all significantly associated with both exposure and outcome variables. Finally, in the association between physical abuse and depressive disorders, only education status and gender were significantly associated with both exposure and outcome– site was not a confounder.

**Table 5. Confounding analyses of ACE-IQ variables with exposure and outcome variables.**

|  | Adversity binary, N=3146 | Adversity frequency, N=3146 |
|  | P-value | P-value |
|---|---|---|
| Any DV | **<.001** | **<.001** |
| Physical Abuse | **<.001** | **.021** |
| Psychological Abuse | **<.001** | **<.001** |
| Sexual Abuse | **<.001** | **<.001** |
| Anxiety Disorders | **<.001** | **<.001** |
| Depressive Disorders | **<.001** | **<.001** |
| Common Mental Disorders | **<.001** | **<.001** |

**Table 6. Confounding analyses of wealth index variables with exposure and outcome variables.**

|  | Wealth Index Quintile | Wealth Index Score |
|  | P-Value | P-Value |
|---|---|---|
| Any DV | .179 | .428 |
| Physical Abuse | **.006** | .123 |
| Psychological Abuse | **.011** | .694 |
| Sexual Abuse | .413 | .260 |
| Anxiety Disorders | .506 | .538 |
| Depressive Disorders | .819 | .700 |
| Common Mental Disorders | .985 | .753 |

**Table 7. Multivariate analyses of any DV with anxiety disorders, adjusted for confounders, bolded if p<0.05.**

|  | Anxiety Disorders | Adjusted Odds Ratio without adversity frequency (95% confidence interval) | Adjusted Odds Ratio with adversity frequency (95% confidence interval) |
|---|---|---|---|
| Any DV | 1.304 | 1.249 (.869, 1.794) | 1.340 (.810, 2.217) |
| Site |  | **1.199 (1.071, 1.342)** | **1.179 (1.028, 1.353)** |
| Education Status |  | **.401 (.221,.728)** | **.398 (.184,.861)** |
| Gender |  | **.484 (.335,.699)** | **.417 (.350,.696)** |
| Adversity Frequency |  |  | **.783 (.669,.917)** |

**Table 8. Multivariate analyses of any DV with depressive disorders, adjusted for confounders, bolded if p<0.05.**

|  | Depressive Disorders | Adjusted Odds Ratio without adversity frequency (95% confidence interval) | Adjusted Odds Ratio with adversity frequency (95% confidence interval) |
|---|---|---|---|
| Any DV | **2.044** | **1.743 (1.135, 2.675)** | .763 (.417, 1.396) |
| Site |  | 1.061 (.933, 1.206) | 1.021 (.869, 1.200) |
| Education Status |  | **.173 (.099,.301)** | .552 (.239, 1.274) |
| Gender |  | **.411 (.260,.650)** | **.409 (.224,.748)** |
| Adversity Frequency |  |  | **.468 (.394,.556)** |

**Table 9. Multivariate analyses of any DV with common mental disorders, adjusted for confounders, bolded if p<0.05.**

|  | Common Mental Disorders | Adjusted Odds Ratio without adversity frequency (95% confidence interval) | Adjusted Odds Ratio with adversity frequency (95% confidence interval) |
|---|---|---|---|
| Any DV | **1.469** | 1.351 (.994, 1.835) | 1.034 (.674, 1.588) |
| Site |  | **1.128 (1.027, 1.238)** | 1.097 (.978, 1.230) |
| Education Status |  | **.276 (.173,.441)** | **.457 (.237,.879)** |
| Gender |  | **.532 (.389,.727)** | **.464 (.301,.714)** |
| Adversity Frequency |  |  | **.626 (.549,.714)** |

## Discussion

The high burden of DV among women, the prevalence of mental disorders among adolescents, and the known negative impacts of DV on maternal mental health and children's overall health in India necessitated further examination of the impact of maternal experiences of DV on adolescent mental health in the Indian context [2,3]. This is further affirmed by studies from around the world that demonstrate links between maternal experiences of DV and the development of mental disorders among adolescents [20]. This study addresses this issue, and stems from the first large-scale interstate data collection on mental health disorders and their correlates in India, and encompasses diverse rural and urban sites including hospitals, schools, and community health centers [42].

The results of the bivariate analysis suggest that maternal experiences of past-year physical, psychological, and sexual abuse are associated with higher rates of anxiety and depression in Indian adolescents. Depressive disorders specifically were associated with physical, psychological, and sexual abuse, while anxiety disorders were associated only with physical and sexual abuse. Psychological violence included verbal threats and harassment, and deprivation of resources, which may have been more noticeable to children in the household. These findings are consistent with studies showing noticeable changes in adolescent behavior and school performance with household violence, indicating they are absorbing its effects [43]. Adversity binary and adversity frequency, site, gender (female or male), and education dropout status (no or yes) were found to be confounders in the association between maternal experiences of DV and adolescent anxiety and depressive disorders, whereas Wealth Index (as reported by quintile and score) was not a confounder. It is not unexpected that adversity binary and adversity frequency are confounders of this association, however, removing the overlapping questions related to household violence in the ACE-IQ (F6-F8) should have reduced redundancy between ACE-IQ and IFVCS. It is not clear, therefore, whether adversity frequency is a true confounder or, rather, a variable that lies along the causal pathway between maternal experiences of DV and adolescent mental health. It could be that maternal experiences of violence lead to increased ACEs, including neglect by parents or displacement of violence onto children, which leads to poorer adolescent mental health outcomes. Gender, additionally, is a relevant confounder, as it suggests that maternal experiences of DV may not be directly impacting female adolescents' mental health, despite increased rates of adverse mental health outcomes among female adolescents. Other factors (e.g., gender norms, substance use in the household), instead, could be influencing the association between maternal experiences of DV and mental health outcomes among female adolescents, existing on causal pathways that may differ from the one under consideration. The increased rates of adverse mental health outcomes in those who had dropped out of school suggest that contextual factors that influence whether adolescents stay in school, including poverty and an increased need for children in the household to earn wages, are related to both household violence and poorer adolescent mental health. These factors have been shown in literature to be varied, with one study showing working can inversely mitigate mental distress for young girls (reduces distress) vs boys (increases distress) exposed to violence [44]. Lastly, site as a confounder could potentially reflect a number of different aspects of study recruitment and design. First, the study sites were very diverse. For example, the Rishi Valley Rural Health Centre is a primary care clinic in a rural area and participants at this site were predominantly recruited from schools in the area, in addition to

children of adult patients of the clinic or children who were patients themselves. NIMHANS, alternatively, is a major mental health hospital in a large urban area, where patients or children of patients were recruited to be study participants. Additional differences in demographics (of both mothers and children) and mental health disorder prevalences amongst participants and their family members could also potentially be reflected in site's ultimate status as a confounder.

The results of the multivariate analysis suggest that physical abuse is correlated with anxiety, depressive disorders, and common mental disorders when controlling for confounders, underscoring the consistent impact that witnessing maternal DV has on adolescents. The odds of having a mother who experienced DV (particularly physical or sexual abuse) were twice as high among adolescents with anxiety, depression or common mental disorders, when controlling for gender, education dropout status, and site. Interestingly, sexual abuse was correlated with anxiety and common mental disorders, although not depressive disorders. While the cause for these differences in association is not clear, they do allude to how types of household violence unwitnessed by children can still impact their mental health, correlating to studies showing exposure (witnessed or unwitnessed) to household violence can alter child stress responses [45]. This association may also be reflected in the pattern noted by study interviewers regarding the progressive nature of abuse (i.e., psychological abuse was more common, and physical and sexual abuse were less common in the initial years of marriage, but physical and sexual abuse were typically associated with a continuation or escalation of violence in the later years of a marriage). Households in which sexual abuse is occurring, therefore, may be households that have had more violence, for more years, with a cumulative effect on the mental health of adolescents in the home.

Overall, the associations between maternal DV and adolescent anxiety, depressive disorders, and common mental disorders parallel studies in Western countries that demonstrate similar associations [46,47]. Our findings serve to corroborate the association between exposure to DV and depression in adolescents demonstrated in other low and middle-income countries [21].

## Limitations

A potential limitation of this study is the four-hour battery of surveys utilized, which may have contributed to participant fatigue and decreased reliability of responses. We also relied on a convenience sampling strategy which reduces the generalizability of our findings. Another notable aspect of the study design is the variability between sites, which yielded a large and diverse sample which we consider a strength of our study; however, different types of sites (e.g., mental health hospitals, primary care centers, university hospitals, schools) may have contributed to site's ultimate determination as a confounding factor in our analysis. Additionally, potential recall bias was introduced when participating mothers were asked about past experiences of DV, physical abuse, psychological abuse, and sexual abuse. We attempted to minimize recall bias by limiting our analysis to experiences of abuse in the past year. Lastly, our analysis did not examine all potential confounders identified in the literature, like family history of mental health disorders and parental substance abuse, as these were not captured in the parent study.

## Conclusions

In conclusion, our findings collectively suggest that maternal DV experience is associated with mental health disorders including anxiety and depression in adolescents in India. Future studies should validate these findings, and independently examine the impact of maternal DV on individual mental disorders, while additionally controlling for parental mental health history, parental history of substance use, parental education level, and site characteristics that were not considered in our analysis. Additionally, control-related variables from the IFVCS scale should be explored in this context to determine their role in understanding the impacts of maternal DV on adolescent mental health. Qualitative studies should investigate the mechanisms by which maternal DV is linked to mental health disorders in paired adolescent children. Further, our findings underscore the need for programming and policy interventions targeted at maternal DV prevention in the context of promoting good mental health among children. Interventions in Indian schools and hospitals to screen for and address adolescent exposure to maternal DV could be employed to mitigate the risk of development of mental health disorders, or

detect and treat existing mental health disorders. Additionally, interventions related to education for parents— elucidating the potential effects of maternal experiences of DV on children's mental health— and teachers— outlining potential signs of mental illness, exposure to maternal DV in the home, and providing training on trauma-sensitive teaching— may be of utility to address the burden of maternal DV exposure on children.

## Appendices

### Physical DV

| | Common Mental Disorders | Adjusted Odds Ratio without adversity frequency (95% confidence interval) | Adjusted Odds Ratio with adversity frequency (95% confidence interval) |
|---|---|---|---|
| Physical DV | **2.339** | **2.095 (1.481, 2.965)** | 1.169 (.717, 1.908) |
| Site | | **1.122 (1.023, 1.231)** | 1.098 (.980, 1.230) |
| Education Status | | **.316 (.197,.507)** | **.466 (.244,.892)** |
| Gender | | **.548 (.403,.745)** | **.503 (.329,.767)** |
| ACEs - Frequency | | | **.637 (.559,.726)** |
| | Anxiety Disorders | Adjusted Odds Ratio without adversity frequency (95% confidence interval) | Adjusted Odds Ratio with adversity frequency (95% confidence interval) |
| Physical DV | **1.904** | **1.809 (1.193, 2.743)** | 1.577 (.897, 2.771) |
| Site | | **1.191 (1.066, 1.331)** | **1.172 (1.024, 1.341)** |
| Education Status | | **.456 (.250,.830)** | **.423 (.197,.908)** |
| Gender | | **.495 (.345,.710)** | **.449 (.272,.740)** |
| ACEs - Frequency | | | **.787 (.673,.922)** |
| | Depressive Disorders | Adjusted Odds Ratio without adversity frequency (95% confidence interval) | Adjusted Odds Ratio with adversity frequency (95% confidence interval) |
| Physical DV | **2.767** | **2.278 (1.418, 3.661)** | .642 (.315, 1.306) |
| Site | | 1.055 (.928, 1.199) | 1.027 (.874,.1.207) |
| Education Status | | **.189 (.108,.332)** | .536 (.234, 1.229) |
| Gender | | **.424 (.270,.667)** | **.428 (.236,.778)** |
| ACEs - Frequency | | | **.469 (.395,.558)** |

### Psychological DV

| | Common Mental Disorders | Adjusted Odds Ratio without adversity frequency (95% confidence interval) | Adjusted Odds Ratio with adversity frequency (95% confidence interval) |
|---|---|---|---|
| Psychological DV | 1.250 | 1.145 (.843, 1.557) | .880 (.572, 1.354) |
| Site | | **1.129 (1.028, 1.240)** | 1.100 (.981, 1.233) |
| Education Status | | **.270 (.170, 1.240)** | **.451 (.234,.869)** |
| Gender | | **.537 (.394,.732)** | **.462 (.300,.711)** |
| ACEs - Frequency | | | **.620 (.545,.706)** |
| | Anxiety Disorders | Adjusted Odds Ratio without adversity frequency (95% confidence interval) | Adjusted Odds Ratio with adversity frequency (95% confidence interval) |
| Psychological DV | 1.129 | 1.073 (.747, 1.541) | 1.092 (.661, 1.802) |
| Site | | **1.199 (1.072, 1.342)** | **1.178 (1.027, 1.350)** |
| Education Status | | **.394 (.218, 1.713)** | **.390 (.180,.843)** |
| Gender | | **.481 (.334,.692)** | **.414 (.248,.690)** |
| ACEs - Frequency | | | **.771 (.661,.901)** |
| | Depressive Disorders | Adjusted Odds Ratio without adversity frequency (95% confidence interval) | Adjusted Odds Ratio with adversity frequency (95% confidence interval) |
| Psychological DV | **1.546** | 1.317 (.857, 2.024) | .647 (.351, 1.193) |
| Site | | 1.063 (.933, 1.210) | 1.027 (.873, 1.208) |

|  | Common Mental Disorders | Adjusted Odds Ratio without adversity frequency (95% confidence interval) | Adjusted Odds Ratio with adversity frequency (95% confidence interval) |
|---|---|---|---|
| Education Status |  | .163 (.094,.283) | .550 (.236, 1.279) |
| Gender |  | .427 (.270,.674) | .414 (.226,.758) |
| ACEs - Frequency |  |  | .465 (.392,.551) |

**Sexual DV**

|  | Common Mental Disorders | Adjusted Odds Ratio without adversity frequency (95% confidence interval) | Adjusted Odds Ratio with adversity frequency (95% confidence interval) |
|---|---|---|---|
| Sexual DV | 2.281 | 1.862 (1.015, 3.415) | .714 (.304, 1.681) |
| Site |  | 1.110 (1.012, 1.217) | 1.098 (.980, 1.229) |
| Education Status |  | .283 (.177,.452) | .468 (.244,.899) |
| Gender |  | .538 (.395,.732) | .499 (.327,.762) |
| ACEs - Frequency |  |  | .619 (.543,.704) |
|  | Anxiety Disorders | Adjusted Odds Ratio without adversity frequency (95% confidence interval) | Adjusted Odds Ratio with adversity frequency (95% confidence interval) |
| Sexual DV | 2.353 | 2.124 (1.087, 4.152) | 1.361 (.557, 3.325) |
| Site |  | 1.176 (1.052, 1.315) | 1.159 (1.012, 1.327) |
| Education Status |  | .421 (.231,.765) | .425 (.197,.921) |
| Gender |  | .490 (.341,.705) | .446 (.270,.737) |
| ACEs - Frequency |  |  | .768 (.656,.898) |
|  | Depressive Disorders | Adjusted Odds Ratio without adversity frequency (95% confidence interval) | Adjusted Odds Ratio with adversity frequency (95% confidence interval) |
| Sexual DV | 2.802 | 2.170 (.982, 4.796) | .366 (.107, 1.248) |
| Site |  | 1.042 (.919, 1.183) | 1.041 (.886, 1.223) |
| Education Status |  | .173 (.099,.302) | .518 (.227, 1.183) |
| Gender |  | .402 (.256,.631) | .431 (.238,.780) |
| ACEs - Frequency |  |  | .469 (.396,.555) |

## Acknowledgments

Thank you to the entire staff at the RVRHC, particularly Geeta, Sujatha, Aruna, and Prasunna, without whom the data collection would not have been possible. Sincere thanks to all the women and children who shared their time and personal experiences for this study. cVEDA consortium members are: Mathew Varghese,[1] Thennarasu Kandavel,[2] Metha Urvakhsh,[1] Satish Girimaji,[3] Preeti Jacob,[3] Deepak Jayarajan,[1] Keshav Kumar,[4] Gitanjali Narayanan,[4] Madhu Khullar,[5] Niranjan Khandelwal,[6] Abhishek Ghosh,[7] Amit,[6] Nainesh Joshi,[5] Ningthoujam Debala Chanu,[8] Fujica M.C.,[8] Victoria Ph,[8] Celina Phurailatpam,[8] Debangana Bhattacharya,[9] Bidisha Haque,[9] Alisha Nagraj,[9] Arpita Ghosh,[9] Anirban Basu,[9] Mriganka Pandit,[9] Subhadip Das,[9] Anupa Yadav,[9] Surajit Das,[9] Sanjit Roy,[9] Pawan Maurya,[9] Geetha Rani T,[10] Sujatha B,[10] Madhavi Rangaswamy,[11] Caroline Fall,[12] Kiran KN,[13] Ramya MC,[13] Chaitra Urs,[13] Santhosh N,[13] Somashekhara R,[13] Divyashree K,[13] Arathi Rao,[14] Poornima R,[14] Saswathika Tripathy,[15] Neha Parashar,[15] Nayana K B,[15] Ashwini Seshadri,[15] Sathish Kumar,[15] Suneela Baligar,[16] Thamodharan Arumugam,[17] Apoorva Safai,[17] Anthony Cyril,[15] Rashmitha,[16] Ashika Roy,[17] Dhanalakshmi D,[15] Shivamma D,[15] Bhavana B R[15]

1. Department of Psychiatry, National Institute of Mental Health and Neurosciences, Bangalore, India
2. Department of Biostatistics, National Institute of Mental Health and Neurosciences, Bangalore, India
3. Department of Child & Adolescent Psychiatry, National Institute of Mental Health and Neurosciences, Bangalore, India

4. Department of Mental Health and Clinical Psychology, National Institute of Mental Health and Neurosciences, Bangalore, India
5. Department of Experimental Medicine, Post Graduate Institute of Medical Education and Research, Chandigarh, India
6. Department of Radiodiagnosis and Imaging, Post Graduate Institute of Medical Education and Research, Chandigarh, India
7. Department of Psychiatry, Post Graduate Institute of Medical Education and Research, Chandigarh, India
8. Regional Institute of Medical Sciences, Imphal, Manipur, India
9. Indian Council for Medical Research - Centre for Ageing and Mental Health, Kolkata, India
10. Rishi Valley Rural Health Centre, Andhra Pradesh, India
11. Department of Psychology, Christ University, Bangalore, India
12. MRC Lifecourse Epidemiology Unit, University of Southampton, United Kingdom
13. Epidemiology Research Unit, CSI Holdsworth Memorial Hospital, Mysore, India
14. Division of Nutrition, St. John's Research Institute, Bangalore, India
15. Centre for Addiction Medicine, National Institute of Mental Health and Neurosciences, Bangalore, India
16. Molecular Genetics Laboratory, National Institute of Mental Health and Neurosciences, Bangalore, India
17. Department of Neuroimaging and Interventional Radiology, National Institute of Mental Health and Neurosciences, Bangalore, India

## Author contributions

**Conceptualization:** Amritha Gourisankar, Ameeta Kalokhe, Rachel Waford Hall, Nilakshi Vaidya, Eesha Sharma, Bharath Holla, Debasish Basu, Rose Dawn Bharath, Amit Chakrabarti, Sylvane Desrivieres, Matthew Hickman, Kamakshi Kartik, Krishnaveni Ghattu, Kumaran Kalyanaraman, Murali Krishna, Rebecca Kuriyan, Pratima Murthy, Dimitri Papadopoulos Orfanos, Meera Purushottam, Sunita Simon Kurpad, Rajkumar Lenin Singh, Roshan Lourembam Singh, Bhagyalakshmi Nanjayya Subodh, Mireille B Toledano, Vivek Benegal, Gunter Schumann, Kartik Kalyanram.

**Data curation:** Amritha Gourisankar, Nilakshi Vaidya, Eesha Sharma, Bharath Holla, Rose Dawn Bharath, Amit Chakrabarti, Vivek Benegal, Gunter Schumann, Kartik Kalyanram.

**Formal analysis:** Amritha Gourisankar, Preethi Ravi, Ameeta Kalokhe, Rachel Waford Hall, Eesha Sharma, Bharath Holla, Vivek Benegal, Gunter Schumann.

**Funding acquisition:** Nilakshi Vaidya, Eesha Sharma, Bharath Holla, Vivek Benegal, Gunter Schumann, Kartik Kalyanram.

**Investigation:** Eesha Sharma, Bharath Holla, Debasish Basu, Rose Dawn Bharath, Amit Chakrabarti, Sylvane Desrivieres, Matthew Hickman, Kamakshi Kartik, Krishnaveni Ghattu, Kumaran Kalyanaraman, Rebecca Kuriyan, Pratima Murthy, Dimitri Papadopoulos Orfanos, Meera Purushottam, Sunita Simon Kurpad, Rajkumar Lenin Singh, Roshan Lourembam Singh, Bhagyalakshmi Nanjayya Subodh, Mireille B Toledano, Vivek Benegal, Gunter Schumann, Kartik Kalyanram.

**Methodology:** Nilakshi Vaidya, Eesha Sharma, Bharath Holla, Gunter Schumann, Kartik Kalyanram.

**Project administration:** Nilakshi Vaidya, Eesha Sharma, Bharath Holla, Kamakshi Kartik, Murali Krishna, Vivek Benegal, Gunter Schumann, Kartik Kalyanram.

**Resources:** Vivek Benegal.

**Supervision:** Ameeta Kalokhe, Rachel Waford Hall, Kamakshi Kartik, Vivek Benegal, Gunter Schumann, Kartik Kalyanram.

**Validation:** Ameeta Kalokhe.

**Writing – original draft:** Amritha Gourisankar, Preethi Ravi, Ameeta Kalokhe.

**Writing – review & editing:** Amritha Gourisankar, Preethi Ravi, Ameeta Kalokhe.

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
