## [Decision Letter · Decision Letter 0]

16 Oct 2024

PONE-D-24-20231Examining the impact of maternal experiences of domestic violence on adolescent mental health disorders in IndiaPLOS ONE

Dear Dr. Gourisankar,

Thank you for submitting your manuscript to PLOS ONE. After careful consideration, we feel that it has merit but does not fully meet PLOS ONE’s publication criteria as it currently stands. Therefore, we invite you to submit a revised version of the manuscript that addresses the points raised during the review process.

We look forward to receiving your revised manuscript.

Kind regards,

Hariom Kumar Solanki, M.D.

Academic Editor

PLOS ONE

Journal Requirements:

2. Thank you for uploading your study's underlying data set. Unfortunately, the repository you have noted in your Data Availability statement does not qualify as an acceptable data repository according to PLOS's standards. At this time, please upload the minimal data set necessary to replicate your study's findings to a stable, public repository (such as figshare or Dryad) and provide us with the relevant URLs, DOIs, or accession numbers that may be used to access these data. For a list of recommended repositories and additional information on PLOS standards for data deposition, please see https://journals.plos.org/plosone/s/recommended-repositories .

3. One of the noted authors is a group or consortium “cVEDA Consortium”. In addition to naming the author group, please list the individual authors and affiliations within this group in the acknowledgments section of your manuscript. Please also indicate clearly a lead author for this group along with a contact email address.

Reviewers' comments:

Reviewer's Responses to Questions

**Comments to the Author**

1. Is the manuscript technically sound, and do the data support the conclusions?

Reviewer #1: Yes

Reviewer #2: Yes

Reviewer #3: Yes

Reviewer #4: Yes

2. Has the statistical analysis been performed appropriately and rigorously? 

Reviewer #1: Yes

Reviewer #2: I Don't Know

Reviewer #3: I Don't Know

Reviewer #4: Yes

3. Have the authors made all data underlying the findings in their manuscript fully available?

Reviewer #1: Yes

Reviewer #2: Yes

Reviewer #3: Yes

Reviewer #4: Yes

4. Is the manuscript presented in an intelligible fashion and written in standard English?

Reviewer #1: Yes

Reviewer #2: No

Reviewer #3: Yes

Reviewer #4: Yes

5. Review Comments to the Author

Reviewer #1: This article is very well written and on a very important topic. Every section of the article is detailed beautifully. However, one suggestion from my side would be, with respect to the way forward would be, that if you could add something that the beneficiary side could implement in their lives to mitigate this problem. You have already explained clearly what the policy makers must do. But some initiatives need to be taken at the beneficiary level as well, in spite of the fact that they are the victims in their entirety. If those could be detailed, I would be really grateful.

Reviewer #2: Dear author,

DV is an important public health concern, but I hope there is room for improvement in your write up. Here are a few suggestions.

kindly define the roles of so many authors for secondary data analysis.

Title: title does not clearly state that you are examining the mental health of children of mothers. From the title it looks like it could be adolescent mothers themselves as child marriage is a possibility in south Asian sub-continent. kindly clearly write mothers and "their" adolescent children or any appropriate term to clarify.

background: you talk about gap in knowledge, but you did not state what gap. kindly provide details of the gap in knowledge regarding association of DV on mothers and mental health issues in relevant children. Relevant literature about local studies should be mentioned. If no Indian studies available then this should be stated and studies from other any countries in the region with similar culture should be mentioned.

Exclusion criteria: exclusion is done among the included sample. Not consenting is not an exclusion as they were never included.

methods:

sampling technique for your secondary analysis is missing. You have mentioned about the sampling technique of original data collection was convenience, but what is the sampling you employed? Did you include all children in cVEDA study aged 12-17 years or randomly selected some of them. Please clarify.

in data collection section it is not clear if the data was collected by the researchers in past or you are referring to some information by the surveyors. Kindly explain clearly how data was collected and how you accessed it. What was the purpose of data collection and if there is any duplicity of findings.

Sample size calculation is missing

results:

table 1 about descriptives is not clear. It is a bit haphazard. It needs to be restructured. You are talking about the details of children or their mothers? kindly clearly state descriptive statistics of children.

“Past-year physical and sexual abuse, but not psychological abuse nor any DV, were significantly associated with anxiety disorders in the paired children (see Table 3). associations were found between adolescent depressive disorders and past-year physical, psychological, sexual abuse, and any maternal DV experience, and between adolescent common mental disorders and past-year maternal physical and sexual abuse and any DV.” Kindly, rephrase to more legible language. Each association should be clearly and specifically written in a scientific language.

Table 3 needs to be better formatted properly.

Discussion: should be further strengthened through relevant literature.

details about informed consent should be further explained.

Reviewer #3: Dear Author,

it is a great effort writing this manuscript. To make it more appealing, I would suggest to reduce the number of tables and re-design table1 specifically. table is very long and row-wise there is not much information.

Reviewer #4: Comments:

1. Section: Methods; Sub-section: Instrument- please operationalize adverse child events

2. Section: Result: Any data from nuclear family?

3. Section: limitation: is there any recall bias when respondents were asked to response against the variable in violence scale (e.g. not in past year but previously in my married life)

6. PLOS authors have the option to publish the peer review history of their article (what does this mean? ). If published, this will include your full peer review and any attached files.

**Do you want your identity to be public for this peer review?** For information about this choice, including consent withdrawal, please see our Privacy Policy .

Reviewer #1: **Yes: ** Dr Yash Alok

Reviewer #2: No

Reviewer #3: No

Reviewer #4: No

---

## [Author Response · Author response to Decision Letter 1]

28 Nov 2024

Dear reviewers and PLOS One editorial team,

We are grateful for your comments and for this opportunity to improve upon our submission. We have included how we have addressed each comment related to the journal’s requirements and reviewers’ comments below. When addressing reviewers’ comments, we have included the original comment and included our responses as bulleted points under each.

Journal Requirements:

Regarding PLOS One’s style requirements: the file naming was edited to be in lower case. Sentence style and the affiliation list was reviewed to ensure consistency with PLOS rules.

Regarding our study’s underlying data set: the cVEDA dataset is maintained on a secure website, available on request, per the protocol outlined on the cVEDA project website: https://cveda-project.org/access-to-the-c-veda-dataset/. The cVEDA study was run by a third party, requiring approvals for us to access the data needed for our study.

Regarding the “cVEDA Consortium” author group: A corresponding author and email address was added. The location of the consortium group full list was added to the affiliations page. The consortium group full list was named in the Acknowledgments section, along with their affiliations.

Regarding the full ethics statement: The full ethics declaration from the published study protocol has been included in the Methods section, including the name of the original study ethics committee, IRB committee for the secondary data analysis, and consent process.

Regarding our reference list: we have added 3 references (48, 49, 50 on the reference list) as we bolstered our discussion. We have otherwise reviewed our reference list to ensure that it is complete and correct.

Reviewers’ comments:

In response to Reviewer #2 answering “No” to the question “Is the manuscript presented in an intelligible fashion and written in standard English?”, we have reviewed the manuscript for typographical and grammatical errors.

Reviewer #1:

This article is very well written and on a very important topic. Every section of the article is detailed beautifully. However, one suggestion from my side would be, with respect to the way forward would be, that if you could add something that the beneficiary side could implement in their lives to mitigate this problem. You have already explained clearly what the policy makers must do. But some initiatives need to be taken at the beneficiary level as well, in spite of the fact that they are the victims in their entirety. If those could be detailed, I would be really grateful.

We appreciate Reviewer #1’s feedback, but are uncertain what is meant by “beneficiary.” We have included information in the Conclusion regarding how findings from this study may be useful in designing interventions for domestic violence in community-based settings including schools and hospitals, which will require a conducive policy environment. Our results will not likely contribute directly to behavior change related to domestic violence (DV), but can be referenced in future efforts targeted at primary prevention.

Reviewer #2:

kindly define the roles of so many authors for secondary data analysis.

We have specifically identified which authors were involved in secondary data analysis (AG, PR).

Title: title does not clearly state that you are examining the mental health of children of mothers. From the title it looks like it could be adolescent mothers themselves as child marriage is a possibility in south Asian sub-continent. kindly clearly write mothers and "their" adolescent children or any appropriate term to clarify.

We have changed the title of the manuscript to Examining the impact of maternal experiences of domestic violence on the mental health of their adolescent children in India to ensure readers understand that the children involved in the study corresponded to the mothers whose experiences of DV were assessed.

background: you talk about gap in knowledge, but you did not state what gap. kindly provide details of the gap in knowledge regarding association of DV on mothers and mental health issues in relevant children. Relevant literature about local studies should be mentioned. If no Indian studies available then this should be stated and studies from other any countries in the region with similar culture should be mentioned.

We specify that the existing gaps in knowledge are related to the impact of maternal experiences of DV on mental health outcomes among their adolescent children, including common mental disorders, externalizing disorders, and suicidality, in the Indian context. We have referenced studies done in India, in addition to studies done in other parts of the world (e.g., Bangladesh, Tanzania, United States) that are related to our research question. We have changed the language in the first paragraph of the introduction to clarify the gap we are seeking to address, in addition to what relevant literature exists.

Exclusion criteria: exclusion is done among the included sample. Not consenting is not an exclusion as they were never included.

We have removed “refusal of consent” as part of the exclusion criteria. Individuals who did not provide consent were not included in the sample, and therefore, could not be subsequently excluded as Reviewer #2 mentions.

methods:

sampling technique for your secondary analysis is missing. You have mentioned about the sampling technique of original data collection was convenience, but what is the sampling you employed? Did you include all children in cVEDA study aged 12-17 years or randomly selected some of them. Please clarify.

In response to this comment, we have clarified that all children aged 12-17 in the original cVEDA study were included in our sample for secondary data analysis.

in data collection section it is not clear if the data was collected by the researchers in past or you are referring to some information by the surveyors. Kindly explain clearly how data was collected and how you accessed it. What was the purpose of data collection and if there is any duplicity of findings.

We have clarified that survey data was collected as part of the original cVEDA study and subsequently used for secondary analysis in our study, after we gained the appropriate approvals from the Executive Committee of the original study.

Sample size calculation is missing

We described our sample size in the Results section (2,784 mother-child pairs), in addition to including it in Table 1. For the purposes of clarity, we have also added this information to the Methods section, based on comments from Reviewer #2.

results:

table 1 about descriptives is not clear. It is a bit haphazard. It needs to be restructured. You are talking about the details of children or their mothers? kindly clearly state descriptive statistics of children.

We are unable to consolidate Table 1 further in order to include all the necessary demographic information; however, we have specified in the caption that the participants are the adolescent children participants.

“Past-year physical and sexual abuse, but not psychological abuse nor any DV, were significantly associated with anxiety disorders in the paired children (see Table 3). associations were found between adolescent depressive disorders and past-year physical, psychological, sexual abuse, and any maternal DV experience, and between adolescent common mental disorders and past-year maternal physical and sexual abuse and any DV.” Kindly, rephrase to more legible language. Each association should be clearly and specifically written in a scientific language.

We have changed the wording of the sentence above summarizing the results included in Table 3 to be more clear and consistent.

Table 3 needs to be better formatted properly.

Table 3 has been reformatted to be more aesthetically consistent and neat.

Discussion: should be further strengthened through relevant literature.

The discussion has been bolstered with inclusion of additional key literature findings that contextualize our results and discussion (including references 48, 49, and 50).

details about informed consent should be further explained.

Additional details regarding the informed consent process (including obtaining informed consent from parents/guardians and assent from children) have been included in the Ethics Declaration section.

Reviewer #3:

Dear Author,

it is a great effort writing this manuscript. To make it more appealing, I would suggest to reduce the number of tables and re-design table1 specifically. table is very long and row-wise there is not much information.

We appreciate Reviewer #3’s comment regarding Table 1, however, we are unable to consolidate Table 1 further in order to include all the necessary demographic information. We have specified in the caption that the participants are the adolescent children.

Reviewer #4:

1. Section: Methods; Sub-section: Instrument- please operationalize adverse child events

With regards to “operationalizing adverse child events,” we have described adverse childhood events (ACEs) in the context of the Adverse Childhood Experiences — International Questionnaire (ACE-IQ) tool, which was used to measure the types and experiences of abuse experienced by children in their households and/or communities. We also included examples of specific ACEs encompassed by this scale.

2. Section: Result: Any data from nuclear family?

In terms of data from the “nuclear family,” we specify that the dataset included in this analysis includes information from mother-child pairs only. Data was not collected from other members of the family; this also falls outside of the scope of examining the impact of maternal experiences of DV on the mental health outcomes of their corresponding adolescent children.

3. Section: limitation: is there any recall bias when respondents were asked to response against the variable in violence scale (e.g. not in past year but previously in my married life)

We appreciate Reviewer #4’s comments about potential recall bias when asking about mothers’ past experiences of abuse. We have addressed this in the Limitations section.

Additional revisions:

While reviewing the manuscript, we have also clarified why and how control-related variables were not included in our analysis. Despite being an important component of the Indian Family Violence and Control Scale, control-related variables were difficult for us to operationalize in this context (i.e., we are unable to assess how children perceive control-related behaviors in the household in a standardized fashion), which we address in the discussion.

Thank you so much for your time and careful consideration in reviewing our work. We appreciate your insight which has enabled us to improve this manuscript, and your affirmations regarding the importance of this work.

Regards,

Amritha Gourisankar, Preethi Ravi, et al.

---

## [Decision Letter · Decision Letter 1]

5 Mar 2025

Examining the impact of maternal experiences of domestic violence on the mental health of their adolescent children in India

PONE-D-24-20231R1

Dear Dr. Gourisankar,

We’re pleased to inform you that your manuscript has been judged scientifically suitable for publication and will be formally accepted for publication once it meets all outstanding technical requirements.

Kind regards,

Hariom Kumar Solanki, M.D.

Academic Editor

PLOS ONE

Additional Editor Comments (optional):

Reviewers' comments:

Reviewer's Responses to Questions

**Comments to the Author**

1. If the authors have adequately addressed your comments raised in a previous round of review and you feel that this manuscript is now acceptable for publication, you may indicate that here to bypass the “Comments to the Author” section, enter your conflict of interest statement in the “Confidential to Editor” section, and submit your "Accept" recommendation.

Reviewer #4: All comments have been addressed

Reviewer #5: All comments have been addressed

2. Is the manuscript technically sound, and do the data support the conclusions?

Reviewer #4: Yes

Reviewer #5: Partly

3. Has the statistical analysis been performed appropriately and rigorously? 

Reviewer #4: N/A

Reviewer #5: No

4. Have the authors made all data underlying the findings in their manuscript fully available?

Reviewer #4: Yes

Reviewer #5: Yes

5. Is the manuscript presented in an intelligible fashion and written in standard English?

Reviewer #4: Yes

Reviewer #5: No

6. Review Comments to the Author

Reviewer #4: The authors addressed all the issues raised in previous revision. They also put their clarifications against other comments made by other reviewers.

Reviewer #5: (No Response)

7. PLOS authors have the option to publish the peer review history of their article (what does this mean? ). If published, this will include your full peer review and any attached files.

**Do you want your identity to be public for this peer review?** For information about this choice, including consent withdrawal, please see our Privacy Policy .

Reviewer #4: No

Reviewer #5: **Yes: ** Mohammad Ashraful Amin

---

## [Editor Report · Acceptance letter]

PONE-D-24-20231R1

PLOS ONE

Dear Dr. Gourisankar,

I'm pleased to inform you that your manuscript has been deemed suitable for publication in PLOS ONE. Congratulations! Your manuscript is now being handed over to our production team.

Kind regards,

on behalf of

Dr. Hariom Kumar Solanki

Academic Editor

PLOS ONE